# Wireless Sensing of Concrete Setting Process

**DOI:** 10.3390/s20205965

**Published:** 2020-10-21

**Authors:** Giselle González-López, Jordi Romeu, Ignasi Cairó, Ignacio Segura, Tai Ikumi, Lluis Jofre-Roca

**Affiliations:** 1School of Telecommunication Engineering, Universitat Politècnica de Catalunya, 08034 Barcelona, Spain; romeu@tsc.upc.edu (J.R.); luis.jofre@upc.edu (L.J.-R.); 2WitekLab, 08910 Badalona, Spain; ignasi@witeklab.com; 3Department of Civil and Environmental Engineering, Universitat Politècnica de Catalunya, 08034 Barcelona, Spain; ignacio.segura@upc.edu (I.S.); tai.ikumi@smartengineeringbcn.com (T.I.); 4Smart Engineering, 08034 Barcelona, Spain

**Keywords:** RFID, concrete setting, relative permittivity, embedded

## Abstract

An RFID-based wireless system to measure the evolution of the setting process of cement-based materials is presented in this paper. The system consists of a wireless RFID temperature sensor that works embedded in concrete, and an external RFID reader that communicates with the embedded sensor to extract the temperature measurement conducted by the embedded sensor. Temperature time evolution is a well known proxy to monitor the setting process of concrete. The RFID sensor consisting of an UWB Bow Tie antenna with central frequency 868 MHz, matched to the EM4325 temperature chip through a T-match structure for embedded operation inside concrete is fully characterized. Results for measurements of the full set up conducted in a real-scenario are provided.

## 1. Introduction

Concrete is the most widely used construction material in the world. Appropriate monitoring of its setting process is necessary to guarantee that it has reached the required structural strength. Having reliable procedures to assess the state of the setting process on-site and in a non-invasive way, can be helpful in speeding up the construction process through an accurate prediction of formwork stripping, post-tensioning, and opening pavement to traffic times. This is also true in the manufacturing of precast concrete structures in which demolding can be done once the setting process is advanced to the point that the structure can self-sustain and as a consequence molds can be reused at a higher pace and production can increase.

In [1], it is shown that the exothermic hydration process associated to the concrete setting causes temperature and also dielectric constant variations of the concrete mixture in the UHF (Ultra High Frequency) band. Monitoring the changes in the dielectric constant of the concrete mixtures has been proposed in [2] in a wired experimental set-up as a way to monitor the setting process. In [3,4], RFID (Radio Frequency Identification) based sensors have been demonstrated to measure chlorine ion concentration and humidity inside concrete as a means to monitor concrete corrosion. The potentiality of wireless coupled sensors has also been shown in [5,6] by embedding inside the concrete a strain sensor electromagnetically coupled to an external antenna, and in [7], RFID technology is used to couple a concrete-embedded strain gauge to an external reader to monitor 3-axis acceleration of concrete structures. Current systems, such as those in [8,9], employ passive RFID sensor T-tags to monitor the setting process of concrete. However, these systems present a poor performance at the beginning of the setting reaction of concrete, where substantial changes in the dielectric properties of the material are produced, as consequence of their probe-like based design.

In this paper a small-size low-cost RFID sensor designed to be embedded into concrete to monitor the setting process is presented. The operation principle is based on the known correlation between temperature peaks that occur in the concrete mixture and the evolution of the hydration process. When cement, water, aggregate, and additives are mixed together, a significant heat increase occurs due to the exothermic process in the reaction between cement and water (hydration reaction), once the strong exothermic process is completed (1–3 h) the temperature decreases and stabilizes providing an estimation of the mechanical strength reached by the concrete sample. Hence, by measuring the temperature evolution of a concrete sample it is possible to assess the maturity and therefore the strength of a concrete sample throughout its setting process. The most significant temperature changes take place during the first 12 h of the reaction [10,11].

The sensor is designed to sample the temperature and record a temperature log that can be wireless read by a commercial RFID reader. The analysis of the temperature evolution provides an indication of the setting process of the concrete. Due to its small size and low cost, sensors can be embedded into the concrete while it is poured. Its data logging features allow an asynchronous reading of the sensors. As shown in [1], in the initial stages of the concrete setting large changes in its dielectric constant and high losses are expected. In consequence, one of major challenges is designing a sensor that is able to properly operate and transmit the measured data embedded in a heavily lossy changing medium. To this end a specific bow-tie broad band antenna conformed to a cylinder has been designed to offer both low-profile integration in the sensor and more stable operation in front of medium changes. The sensor is based on the EM4325 Integrated Circuit that provides the RFID front end, data memory, and integrated temperature sensor.

The paper is organized as follows: Section 2 contains a detailed description of the RFID-based wireless system for temperature monitoring, including details on the design of the RFID sensor for embedded operation at 868 MHz and the RFID reader, as well as analytical results for the propagation loss in the considered scenario. In Section 3, results corresponding to the in-field measurement have been included.

## 2. RFID-Based System for Wireless Temperature Monitoring

The RFID wireless system to monitor the internal temperature of concrete sample during the setting reaction is structured as depicted in the block diagram in Figure 1. During the pouring stage of concrete, the RFID sensors consisting of an EM4325-IC (Integrated Circuit) temperature sensor matched to a bow-tie broad band antenna and conformed around a 3D printed PLA structure with a cylindrical shape, are poured inside the concrete mixture as part of the aggregates. A commercial RFID reader formed by a circularly polarized patch antenna and a reader ID eNUR-10W from Nordic is placed outside the concrete. This RFID reader is software controlled to make systematic readings of the temperature registered by the wireless sensors previously placed inside the concrete sample. A detailed description of the main design features of the RFID sensor and reader, as well as an analytic study of the propagation path between the external antenna and embedded device, are provided in this section. The block diagram in Figure 1 illustrates the operation principle of the RFID system. Further technical information of the EM4325 chip and the NORDIC ID eNUR-10W reader can be found in [12,13], respectively.

### 2.1. Wireless RFID Temperature Sensor

When designing a RFID sensor to work embedded in a medium whose properties change over time, there are certain constrains that must be taken into account as they will have an impact on the performance of the device that cannot be neglected.

From an industrial point of view, it is desired to come up with a design that is as compact and small as possible. In the construction industry for example, robustness is a must, as a device embedded in a block of concrete has to withstand a tremendous amount of weight and pressure. Taking this into account, a thin planar design is quickly off the table, and a shift towards a more “rock-like” device is preferred as it mimics the shape of aggregates already present in concrete mixtures. A considerable amount of water will be present at the first stages of the setting process of concrete, so it is important to assure that the device is waterproof to prevent damage on the electronics. Nevertheless, the main goal of this device is to monitor the temperature of the surrounding medium, which implies that the sensor must be exposed to the medium somehow for accurate temperature measurement. RFID tags, typically operate as passive devices, which means that they are powered by the RF energy transmitted from a reader. However when they incorporate “sensing” capabilities, working in battery operated mode generally provides more processing capabilities and higher reading range and reliability. The RFID sensor layout must be then designed in a way that simultaneously ensures compactness and robustness, isolation, and an accurate temperature measurement.

The EM4325 RFID chip is a Radio Frequency Identification Integrated Circuit from EM Microelectronic that can be either battery powered or beam powered by RF (Radio Frequency) energy transmitted from a reader. In a Battery Assisted Passive (BAP) configuration, the EM4325 chip offers better sensitivity (−30 dBm) that translates into a superior reading range, compared to purely passive operation. When designing sensors intended to operate inside of lossy materials, as it is the case of concrete, maximizing the Dynamic Range (DR) between the transmitting and receiving antenna is of great interest to cope with the high propagation losses produced as a consequence of concrete’s attenuation. This can be either achieved by increasing the transmission power, or as done in this case, by improving the sensitivity of the embedded passive device. The integrated temperature sensor supports temperatures in the range of −40 ${}^{\circ }$C to +60 ${}^{\circ }$C, which widely covers typical concrete temperatures reached during placement [12].

The EM4325 RFID chip does not include the capability of storing temperature measurements. Meaning that these should either be acquired on demand, or by installing a micro-controller. For the current application, the chip has been configured to operate as a radio-frequency front-end and to perform on demand temperature readings when requested by the external reader.

This chip, when configured in BAP operation mode, has an input impedance of 6.5−j172Ω. In accordance, the RFID antenna conceived to operate matched to the EM4325 chip, is designed to provide a 6.5+j172Ω impedance at its input. To provide the required low resistance and high reactance input impedance, a T-match structure is employed between the bow-tie antenna and RFID chip [14,15].

In order to provide the aforementioned required compactness and robustness, the proposed design is printed on top of a bendable substrate of 0.125 mm thickness from Rogers ($\varepsilon_{rRT5880}=2.2$). This is then wrapped around the external walls of a semi-holed 3D printed PLA cylinder with a diamter of 30 mm. To provide a planar section to place the RFID chip and for isolation purposes, a vertical slice is made on the wall of the cylinder where both the EM4325 RFID chip and the T-match section of the antenna are placed. The removed section is then placed back on top, covering the electronics and isolating it from the surrounding medium. In this way, we are able to protect the electronics from the high water concentration present in the concrete mixture at the beginning of the setting reaction. An additional advantage of placing back on top of the T-match the removed PLA section, is that it enables a reduction of the T-match dimensions, as PLA’s relative permittivity is higher than air’s ($\varepsilon_{rPLA}=2.5$), hence contributing to the compactness of the sensor. A 3V battery is placed inside the PLA cylinder, connected to the RFID chip through biasing lines, for BAP operation. The proposed RFID sensor is represented in Figure 2.

An important factor when assessing the performance of the antenna is the Impedance Coupling Factor (ICF), computed as:(1)$$ICF=\frac{4Re\left(Z_{a}\right)Re\left(Z_{chip}\right)}{{\left(Re\left(Z_{a}\right)+Re\left(Z_{chip}\right)\right)}^{2}+{\left(Im\left(Z_{a}\right)+Im\left(Z_{chip}\right)\right)}^{2}}.$$
where $Z_{a}$ is the antenna impedance and $Z_{chip}$ is the impedance of the RFID chip, 6.5−j172Ω in this case. The Impedance Coupling Factor measures the fraction of power that is effectively transferred from the antenna to the RFID chip (and vice versa), and the design goal is to obtain a value close to one.

Figure 3 represents the the behavior of the proposed T-match bow-tie antenna in terms of input impedance and Coupling Factor, when placed in the embedding scenario. These results have been obtained through EM (Electromagnetic) simulations using CST Microwave Studio, considering the nominal value for the embedding medium’s relative permittivity $\varepsilon_{r}=6-j0.36$ at 0.868 GHz.

It is known from [1] that the real part of the relative permittivity of concrete throughout the setting reaction varies exhibiting maximum values around $\varepsilon_{r}^{\prime }=18$ at the beginning of the reaction and lowering to $\varepsilon_{r}^{\prime }=11$ by the eighth day of the process. With this phenomenon in mind, the Impedance Coupling Factor between the wrapped T-match bow-tie antenna and the RFID-IC has been evaluated for real part relative permittivities ranging between 4 and 18 (Figure 4).

Figure 4 shows that at the operation frequency of 868 MHz, despite the changes produced in the sensor’s antenna impedance as a consequence of the permittivity variations in the embedding medium, the Impedance Coupling Factor is above 0.4 for the expected dielectric permittivity evolution of the concrete sample throughout its setting process.

To enable the temperature sensing capability of the chip, a small hole is drilled on the section of the cylinder that acts as a “cover” for the T-match and chip. This hole is placed right on top of the chip and has dimensions equivalent to those of the sensor. Then a small aluminum cylinder is placed right on top of the temperature sensor (Figure 5b). One face of the aluminum cylinder is left exposed to the external medium while the other one is laying on the surface of the chip where a thermal conductive paste is applied in advance. The walls of this aluminum cylinder are glued to the cover to ensure fixation. Aluminum’s good thermal conductivity makes it a suitable choice to act as an interface between the temperature sensor and embedding medium.

To assure that the whole device is completely isolated from the surrounding medium, the sensor is dipped in a liquid solution of PlastiDip and an air-dry rubber solution is applied to provide at the same time for protection and isolation to the RFID sensor. Figure 5 displays the final manufactured prototype. The main technical features of the proposed RFID temperature sensor are summarized in Table 1.

### 2.2. RFID Reader

The Nordic ID eNUR-10W reader Figure 6a is one of Nordic ID modules for embedded RFID solutions. It is a cost-effective alternative to both fixed and high-end mobile RFID readers with multiple antennas. It is available as a Surface-Mount Device (SMD) component, or mounted on a PCB (Printed Circuit Board) for easier integration. The frequency of operation ranges from 865.6 MHz to 867.6 MHz in Europe, complying with the regulations from the European Telecommunications Standards Institute (ETSI). Regarding power consumption, this reader has an operating power of 4.3 W, and the typical supply voltage is 3.6 VDC, even though it admits voltages from 3.4 to 5.5 VDC at its input. The output power is 30 dBm, and it is adjustable in 1dB steps [13].

The patch antenna connected to the Nordic eNUR reader has a 9 dBi gain and a bandwidth ranging from 840 MHz to 960 MHz (Figure 6b). It has circular polarization, which introduces a loss due to a polarization mismatch of 3 dB as the antenna of the RFID tag has linear polarization. However, due to the random placement of the sensor during the pouring process using a circular polarized reader antenna helps prevent miss-detection due to high polarization losses that could occur in the case of using a linear polarized antenna.

### 2.3. Monitoring Software

The software in charge of establishing, controlling, and monitoring communication between the reader and RFID sensor has been developed by WiTekLab following Nordic ID semiconductors’ recommendations for the development of software solutions for their RFID Embedded Reader Modules.

When the reader is connected to the computer and powered on, the control software scans for tags within the reading distance of the RFD reader, and displays parameters such as tag configuration, sensor data, and current temperature. As the EM4325 sensor does not have the capability of storing temperature measurements, the software tracks the temperature readings of a given tag, and stores it on a CSV file in the computer.

### 2.4. Propagation Loss between the RFID Reader and the Embedded RFID Sensor

The propagation loss between a RFID sensor located inside a concrete structure, and a RFID reader placed outside is represented through Equation (Equation 2). This expression, extracted from [16], has been conceived from the classical approximation for free-space spherical wave propagation, considering that there are several propagation media with relative permitivities much higher than air in the propagation path between the receiving and transmitting antenna:(2)$$\frac{P_{R}}{P_{T}}=-20\log\left(\frac{4\pi d_{air}}{\lambda_{air}}\right)-20\log\left(\frac{4\pi d_{MUT}}{\lambda_{MUT}}\right)-L_{MUT}+e_{r}-L_{CP}$$

In (Equation 2), the first two terms account for the free-space spherical wave propagation in air ($L_{0}\left(dB\right)$) and inside the embedding medium (concrete) ($L_{0MUT}\left(dB\right)$), respectively. $L_{MUT}$ represents losses due to the attenuation of concrete, $e_{r}$ is the reflection efficiency (computed according to [14]) and it accounts for the losses at the interface between the mediums, and $L_{CP}$ are the circular to linear polarization losses.

We have extracted from [1] the relative permittivity values measured at 868 MHz for a concrete sample at days 0.5, 1, 2, 4, and 8 of the setting reaction. Table 2 and Table 3 summarize all communication link parameters, while Figure 7 represents the propagation loss in the considered scenario over distance $d_{air}$ computed through Equation (Equation 2). The radiation efficiency ($e_{rad}$) in Table 2, has been obtained from the simulations conducted in CST of the proposed RFID sensor. This parameter represents the ratio of the power radiated over the total input power [14]. The parameter $L_{TMUT}$(dB) in Table 3 accounts for the total propagation loss in the embedding medium ($L_{MUT}$(dB) + $L_{0MUT}$(dB)).

From Table 2, we can extract that the dynamic range of the current scenario, considering the transmission power of the RFID reader ($P_{T}$), the sensitivity of the RFID sensor ($P_{R}$), and the gain of both transmitting and receiving antennas, is of −68dB. The analytical results presented in Figure 7 provide an estimation of the maximum distance where the external RFID reader can be placed with respect to the interface with the concrete medium, considering that the maximum propagation loss would correspond to the available dynamic range. Hence, at the beginning of the setting reaction, the reader should be placed no further than 10 cm from the medium interface, while at the eighth day of setting the mixture has dried enough that it is possible to place the reader over 1 m away from the embedding medium’s interface.

## 3. In-Field Measurements

The RFID temperature sensor characterized in the previous section (Figure 5), has been tested in a real life scenario at PROMSA’s (one of Spain’s major concrete manufacturers) testing facilities. These facilities have been conceived to conduct controlled concrete manufacturing and sampling at room temperature. The concrete mixture employed in the measurement was prepared in the moment, tested for a previously predetermined consistency, and poured in a series of wooden containers of dimensions 30×30×30 cm and 2 cm thickness. The concrete mixture was the same as the one corresponding to the relative permittivity values mentioned in Table 3.

Figure 8 depicts the scenario of the in-field measurement. On the left (Figure 8a), it is possible to observe the RFID sensors before being fully convered by the concrete, placed one vertically and the other horizontally, to asses the viability of having linearly polarized embedded antennas, and a circularly polarized external antenna. The green cables are thermocouples inserted right next to the position of the sensors and attached to a data logger, to later compare the temperature reading made by each one of the sensors and correlate it to the one registered by the corresponding thermocouple. The whole layout of the measurement scenario can be observed in Figure 8b. The Nordic ID eNUR RFID reader is software controlled through a computer and it is connected to the patch antennas in the background, which are aimed towards the wooden containers where the RFID sensors are to be embedded.

A total of four sensors were placed inside two wooden containers. Two sensors were inserted at the center of each container (s1 and s2, $d_{MUT}$ = 0.15 m), another one was placed at mid-height next to one corner (s3, $d_{MUT}$ = 0.02 m), and the last one at the top corner of one of the containers (s4, $d_{MUT}$ = 0.02 m).

At the beginning of the measurement, it was only possible to detect the sensor placed on the corner of the container (s4), by placing the reader’s antenna right next to the interface with the container. The input power received at the time from this sensor was −62 dBm. Twenty-four hours later, the sensor located at mid-height next to the corner (s3) was also detected. At this time, the input power received from this sensor was −60 dBm, while the one received from s4 had risen to −50 dBm.

The sensors located at the center of the concrete containers (s1 and s2) were not detected during this test. We were able to retrieve them after concrete compression tests had been conducted, and it was possible to find out that the concrete’s chemical corrosive effect had in fact damaged the isolating rubber covering the sensors, water had then leaked to the electronics and permanently damaged the sensor.

The results exhibited in Figure 9 correspond to the temperature measured by sensor s4, compared to the temperature measurement made by the corresponding thermocouple. The temperature measured by an additional thermocouple set to register the external temperature has also been included in Figure 9. From the results in Figure 9, we are able to say that the temperature measurement made by our embedded wireless RFID temperature sensor is a faithful representation of the actual temperature curve produced during the setting reaction of concrete. It is possible to notice that the oscillations present in all three temperature curves correspond to the variations produced during the day-night temperature cycle, as they happened in a 24 h period. In the top right figure within Figure 9, the differential temperature for both the thermocouple and the RFID sensor placed inside the concrete sample have been represented. This is done by subtracting the room temperature effect, therefore mitigating the aforementioned day-night temperature cycle.

The results of the peak temperature values achieved during the first hours of the hydration reaction in Figure 9, are in good agreement with those presented by [8,9] obtained using semi-embedded T-tag RFID sensors and also validated using thermocouples. The presented RFID sensor has also proven to remain well matched throughout the variations produced in the dielectric properties of the medium (validating the results presented in Figure 4 for the antenna Impedance Coupling Factor) and not interfering with the evolution of the setting process as it is fully wireless.

The good agreement between the temperature read obtained with the thermocouples and that obtained with the RFID sensor, and the fact that it was possible to maintain a stable communication link between the embedded sensor and the external reader, are proof of the feasibility of transmitting a microwave signal in this kind of scenario without it being interfered by the attenuation and changing dielectric properties inherent of cement-based materials. These results are in agreement with those presented in [17], where the anti-interference performance of RFID systems has been studied for soil-, water-, and metal-covered sensors. Additionally, network based communication methods as those proposed in [18,19,20], can be a suitable solution in high interference scenarios.

Once validated, the feasibility of the proposed RFID system for wireless monitoring of the setting reaction produced in concrete over time at 0.868 GHz, it could be of interest for potential designers to compare the attenuation parameter $L_{TMUT}$ (see Table 3) obtained from the results presented in [1], at 0.43 GHz, 0.868 GHz (this paper), and 2.45 GHz. These are typical frequencies in the ISM band employed in sensing applications. Table 4 displays the total propagation loss in the embedding medium ($L_{TMUT}$) for the three considered frequencies at several time instants along the setting reaction. From these results, it may be concluded that for the same communication link parameters (transmitted power, sensitivity, and antenna gain), the selected operation frequency is a critical parameter to take into account when designing RFID systems to operate embedded in high permittivity lossy materials. For the presented scenario, and in terms of the propagation loss, RFID solutions operating at frequencies below 0.868 GHz (e.g., 0.43 GHz) may be achievable, while those having an operation frequency above 0.868 GHz (e.g., 2.45 GHz) may not be considered feasible. When size constrains are taken into consideration, 0.868 GHz seems to be a reasonable choice for these applications.

## 4. Conclusions

The results presented in this manuscript are proof of the possibility of employing wireless technology to remotely monitor the physical and chemical processes produced during the setting reaction of concrete using temperature measurements as a proxy for the evolution of a hydration process that occurs during the setting of concrete. This information can be of use at construction sites to reduce construction time, as well as for safety purposes such as building’s structural strength monitoring.

The reading distances obtained during in-field measurements were in line with the propagation attenuation values employed for propagation loss over distance in air estimation. This means that available data to describe this communication scenario accurately represents the real propagation conditions.

The results presented in this paper for the proposed RFID sensor validate its adequacy for remote wireless monitoring of the hydration reaction of concrete during its curing process by means of a stable temperature measurement conducted with a fully-embedded wireless RFID sensor. The wide band capabilities of the presented device make it possible to set up a precise and stable communication link with an external reader from the beginning of the hydration reaction, which validates that the sensor remains well matched despite the change produced in the dielectric properties of a concrete mixture throughout the setting reaction.

This system using active RFID technology can be easily adapted to integrate further sensors to the electronics of the embedded device (e.g., moisture sensor) and conduct a broader characterization of the physical and chemical properties of the concrete sample. By tuning the energy consumption of the RFID device, it is also possible to conduct long term monitoring of the medium. The same wireless system can also be used in other construction materials such as self-leveling mortar for similar purposes.

## 5. Patents

The patent entitled “SISTEMA DE TRANSMISIÓN ENTRE UN MEDIO DE TRABAJO Y UN MEDIO DE DESTINO CON DIFERENTES PERMITIVIDADES” with application number P202030087 is currently being processed.

## Figures and Tables

**Figure 1 sensors-20-05965-f001:**
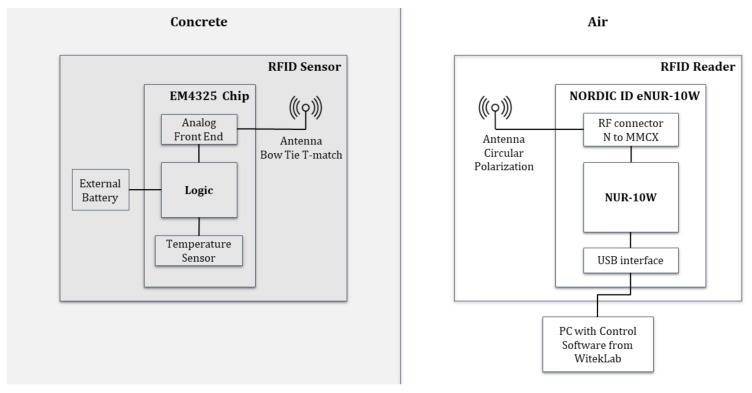
Block Diagram of the RFID-based system for wireless temperature monitoring.

**Figure 2 sensors-20-05965-f002:**
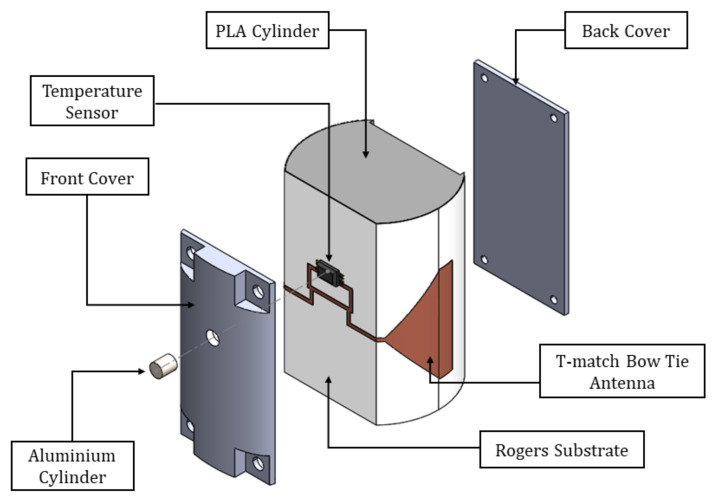
Wrapped RFID antenna. Bow tie with T-match.

**Figure 3 sensors-20-05965-f003:**
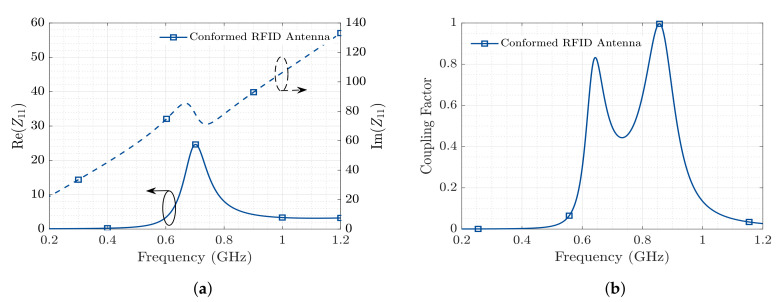
Antenna parameters: (**a**) Input impedance. Re(Z_11_) left vertical axis and Im(Z_11_) right vertical
axis and (**b**) impedance coupling factor.

**Figure 4 sensors-20-05965-f004:**
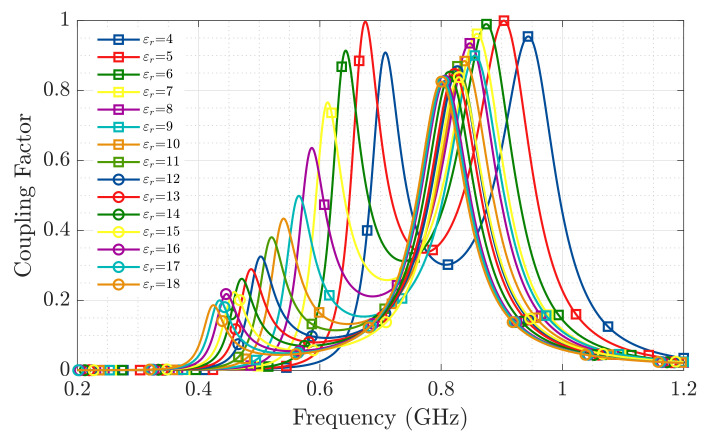
Antenna impedance coupling factor between the wrapped bow tie T-match antenna and the RFID-IC for $\varepsilon_{r}$ between 4 and 18.

**Figure 5 sensors-20-05965-f005:**
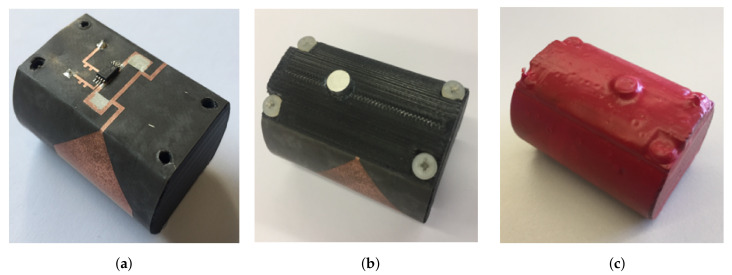
Manufactured RFID Temperature Sensor: (**a**) RFID sensor without cover; (**b**) assembled
sensor; and (**c**) sensor after protection layer.

**Figure 6 sensors-20-05965-f006:**
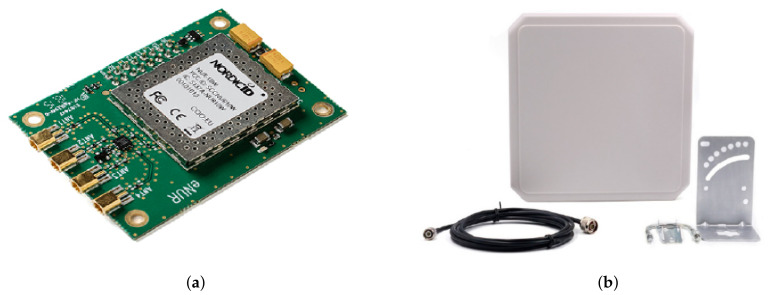
RFID Reader: (**a**) Nordic ID eNUR RFID reader and (**b**) antenna of the eNUR reader.

**Figure 7 sensors-20-05965-f007:**
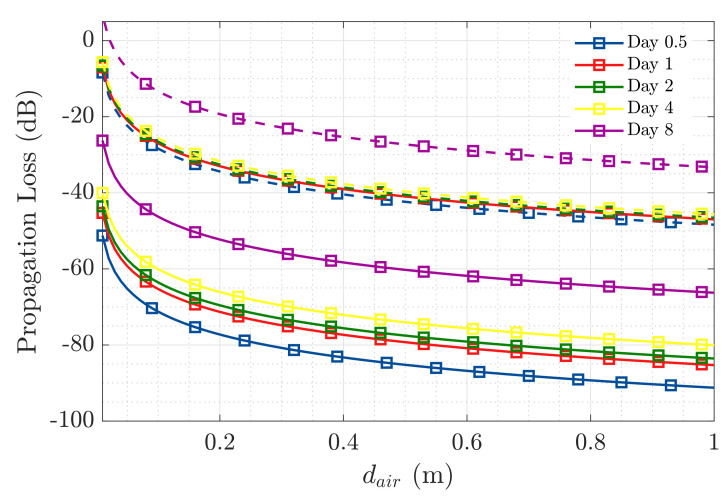
Dynamic range over $d_{air}$ at 868 MHz, for days 0.5, 1, 2, 4, and 8 of concrete’s setting reaction, for a sensor embedded 0.15 m ($d_{MUT}$) inside concrete (straight lines), and one embedded 0.02 m (dashed lines).

**Figure 8 sensors-20-05965-f008:**
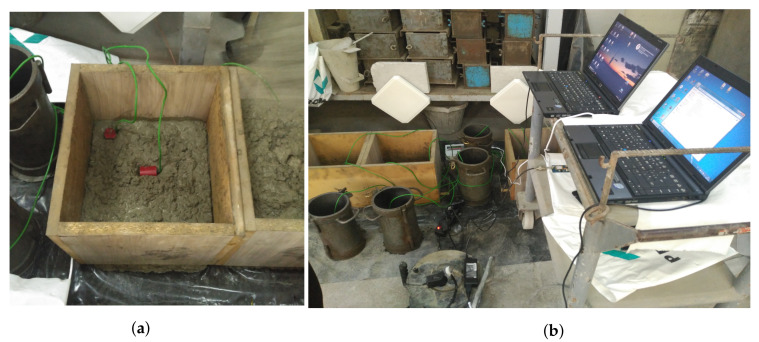
In-field measurement at PROMSA: (**a**) Location of the embedded RFID sensors and (**b**) layout
of the whole measurement scenario.

**Figure 9 sensors-20-05965-f009:**
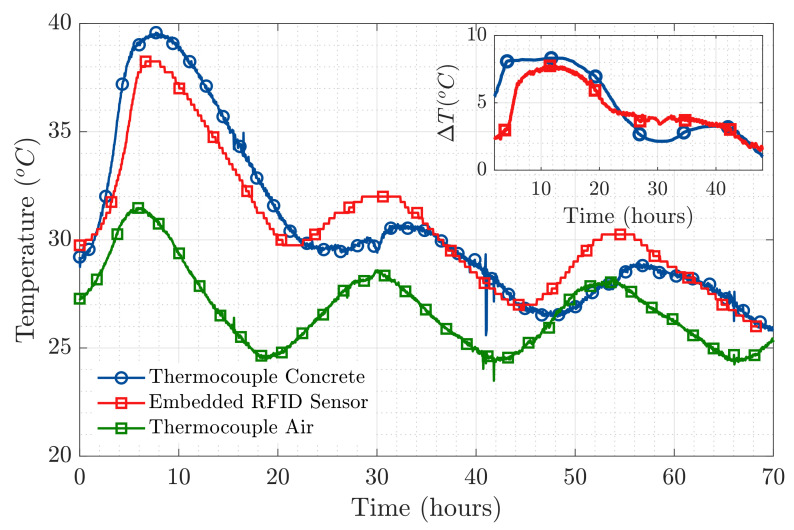
Main Figure: Temperature measured by the RFID sensor compared to temperature from the thermocouples. Top right Figure: Differential temperature inside concrete, removing the effect of room temperature.

**Table 1 sensors-20-05965-t001:** Technical features of the RFID temperature sensor.

Parameter	Value	Description
**Dimensions**	41.2 mm × 30 mm	Height × Diameter of the RFID sensor.
**Volume**	0.0172 m${}^{3}$	Volume of the RFID sensor.
**Weight**	23 g	Weight of the RFID sensor.
**Temperature range**	−40 ${}^{\circ }$C to +60 ${}^{\circ }$C	Temperature range of the RFID sensor.
**Temperature Accuracy**	±1 ${}^{\circ }$C	Temperature accuracy of the EM4325 chip.
***Z_EM4325_***	6.5−j172Ω	Input impedance of the EM4325 chip in BAP mode.
**Antenna Radiation Pattern**	Dipole-like (Omnidirectional)	Shape of the Bow Tie T-match antenna radiation pattern in azimuth.
**Device Consumption**	15.9 μW	Power consumption per reading of the EM4325 RFD chip.

**Table 2 sensors-20-05965-t002:** RFID communication link parameters.

Parameter	Value	Description
$P_{R}$	−31 dBm	Sensitivity of the EM4325 chip.
$P_{T}$	30 dBm	Output Power of the Nordic eNUR reader.
$G_{T}$	9 dBi	Gain of the RFID Reader Antenna.
$D_{R}$	2.95 dBi	Directivity of the RFID Sensor Antenna.
$e_{rad}$	0.4	Radiation Efficiency of the RFID Sensor Antenna.
$d_{MUT}$	0.15 m, 0.02 m	Propagation distance inside concrete medium.
$\lambda_{Air}$	0.345 m	Wavelength in Free Space ($\varepsilon_{r}=1$).
$L_{CP}$	3 dB	Polarization Missmatch Losses.

**Table 3 sensors-20-05965-t003:** Concrete propagation parameters for $d_{MUT}$ = 0.15 m and 0.02 m

	$\varepsilon_{r}$	$\lambda_{r}$ (m)	$\alpha_{\mathrm{MUT}}$ (dB/cm)	$L_{\mathrm{MUT}}$ (dB) (0.15 m/0.02 m)	$L_{0\mathrm{MUT}}$ (dB) (0.15 m/0.02 m)	$L_{T\mathrm{MUT}}$ (dB) (0.15 m/0.02 m)	$e_{r}$ (dB)
Day 0.5	16.8 − j10	0.084	1.95	29.25/3.9	27.02/9.52	56.27/13.42	−6.73
Day 1	15.5 − j9.8	0.088	1.6	24/3.2	26.62/9.12	50.62/12.32	−6.43
Day 2	15 − j8.9	0.089	1.5	22.5/3	26.52/9.02	49.02/12.02	−6.3
Day 4	14.2 − j6.2	0.092	1.3	19.5/2.6	26.23/8.73	45.73/11.33	−6.09
Day 8	13.7 − j5.7	0.093	1.19	17.85/2.38	26.14/8.64	43.99/11.02	−5.96

**Table 4 sensors-20-05965-t004:** Concrete propagation parameters for $d_{MUT}$ = 0.15 m and 0.02 m at 0.43 GHz, 0.868 GHz, and 2.45 GHz.

$L_{T\mathrm{MUT}}$(dB) @(0.15 m/0.02 m)	f = 0.43 GHz	f = 0.868 GHz	f = 2.45 GHz
**Day 0.5**	44.24/6.77	56.27/13.42	82.8/24.62
**Day 1**	39.14/5.93	50.62/12.32	76.67/23.37
**Day 2**	36.23/5.55	49.02/12.02	78.44/23.3
**Day 4**	33.25/4.99	45.73/11.33	77.28/23
**Day 8**	31.79/4.67	43.99/11.02	78.06/23.02

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
