# Peer review of "Wireless Sensing of Concrete Setting Process"

_sensors, 2020, doi:10.3390/s20205965_

Round 1
Reviewer 1 Report
This paper presents a RFID-based wireless system to measure the evolution of the setting process of cement-based materials. The organization of tha paper is well structured and the experimental results seem convincing enough. The pratical background is strong. Before acceptance, some improvements should be made first. 1) The authors need to illustrate the related work regarding the wireless sensing of concrete setting process. For example, the limitations of existing systems in this task. 2) In the experiment, the authors only evaluate the accuracy of the proposed system. In my opionion, the anti-interference performance of the wireless system is my major concern. So how about the anti-interference of this system. 3) To highlight the scientific value of this work, it is necessary for authors to design more experiments. 4) In my opinion, the wireless RFID systems is a typical IoTs system. In the introduction part, some references about IoTs should be added: "Message forwarding for WSN-Assisted Opportunistic Network in disaster scenarios", "Cascading failures in wireless sensor networks with load redistribution of links and nodes" and "Modeling cascading failures for wireless sensor networks with node and link capacity".Author Response
Please see the attachment.

Reviewer 2 Report
The paper reports a contactless temperature sensor for monitoring the concrete setting process.
The sensor is built from off-the-shelf circuits and devices.
The antenna is customized.
A few comments below:
- The authors give little detail about technical aspects of the sensor
- Detailed explanations about the setting x temperature could improve the paper.
- In Fig. 10 the authors show the temperature measured. Isnt't it necessary to subtract the room temperature from the concrete temperature?
- The authors should compare their results with the state-of-the-art literature.
- More results are necessary
- References that can be added:
- Liu Y, Deng F, He Y, Li B, Liang Z, Zhou S. Novel Concrete Temperature Monitoring Method Based on an Embedded Passive RFID Sensor Tag. Sensors (Basel). 2017;17(7):1463. Published 2017 Jun 22. doi:10.3390/s17071463
- S. Manzari, T. Musa, M. Randazzo, Z. Rinaldi, A. Meda and G. Marrocco, "A passive temperature radio-sensor for concrete maturation monitoring," 2014 IEEE RFID Technology and Applications Conference (RFID-TA), Tampere, 2014, pp. 121-126, doi: 10.1109/RFID-TA.2014.6934212.
Reviewer 3 Report
Paper can be accepted after the following corrections:
- Please provide overall schematic block diagram of sensing module and receiving station.
- Figure's captions should be presented accordingly to Publisher's requirements.
- Bibliography should be developed. There is large number of previously developed systems focused on RFID-based concrete temperature monitoring.
- Please provide more detailed information about testing methodology at PROMSA
- Figure 7 is not suitable for the scientific publication.
- Conclusion should be developed to present the most important achievements considering the state-of-the-art.
Round 2
Reviewer 1 Report
qualified for acceptance.
Reviewer 2 Report
I congratulate the authors for the improvements provided.
All the comments and questions were adequatelly addressed and the paper can be accepted in its current form.
Reviewer 3 Report
Paper was corrected and can be accepted in the present form.